# Synthesis of Novel Pyrazole Derivatives Containing Phenylpyridine Moieties with Herbicidal Activity

**DOI:** 10.3390/molecules27196274

**Published:** 2022-09-23

**Authors:** Zengfei Cai, Wenliang Zhang, Zhongjie Yan, Xiaohua Du

**Affiliations:** 1Catalytic Hydrogenation Research Center, Zhejiang Key Laboratory of Green Pesticides and Cleaner Production Technology, Zhejiang Green Pesticide Collaborative Innovation Center, Zhejiang University of Technology, Hangzhou 310014, China; 2Agrowin (Ningbo) Bioscience Co., Ltd., Ningbo 315100, China

**Keywords:** synthesis, pyrazole, phenylpyridine, herbicidal activity

## Abstract

To discover new compounds with favorable herbicidal activity, a range of phenylpyridine moiety-containing pyrazole derivatives were designed, synthesized, and identified via NMR and HRMS. Their herbicidal activities against six species of weeds were evaluated in a greenhouse via both pre- and post-emergence treatments at 150 g a.i./hm^2^. The bioassay revealed that a few compounds exhibited moderate herbicidal activities against *Digitaria sanguinalis*, *Abutilon theophrasti*, and *Setaria viridis* in post-emergence treatment. For instance, compounds **6a** and **6c** demonstrated 50% inhibition activity against *Setaria viridis*, which was slightly superior to pyroxasulfone. Thus, compounds **6a** and **6c** may serve as the new possible leading compounds for the discovery of post-emergence herbicides.

## 1. Introduction

Pyrazole-containing compounds, a class of five-membered heterocyclic compounds with simple synthetic routes, have been widely used in the study of biologically active molecules such as in medicine [1,2], pesticides [3,4], and veterinary drugs [5,6]. In the field of agriculture, a variety of small molecules containing pyrazole groups have been developed as pesticide products (Figure 1), such as fungicides [7,8], insecticides [9,10,11,12], and herbicides [13,14,15]. Zhang et al. demonstrated that a series of novel substituted pyrazole aminopropyl isothiocyanates exhibited certain herbicidal activity against *Echinochloa crusgalli*, *Cyperus iria*, *Dactylis glomerata*, and *Trifolium repens* [16]. A class of 1-acyl-3-phenyl-pyrazol benzophenones was prepared by Ye et al. using dimethylformamide dimethyl acetal and 1, 3-diphenylpropane-1, 3-dione as the starting materials, which showed good herbicidal activity [17]. Liu et al. reported that a class of novel pyrazole aromatic ketone derivatives exhibited excellent herbicidal activity against various broadleaf weeds treated post-emergence [18]. The promising pesticide pyroxasulfone [19] discovered by Kumiai Chemical is a pre-emergence herbicide that could provide excellent control of grass and broadleaf weeds in corn and soybean fields.

Substituted phenylpyridines discovered by Schaefer et al. exhibited good inhibition activity against weeds [20,21]. Substituted 3-(pyridin-2-yl)benzenesulfonamide derivatives disclosed by Liu et al. showed excellent inhibitory activity against a variety of weeds [22,23,24]. Du et al. also reported that a range of kresoxim-methyl derivatives containing phenylpyridine moieties exhibited higher inhibitory activities against broadleaf weeds than mesotrione [25,26].

Herein, 10 novel pyrazole derivatives containing phenylpyridine moieties were obtained via the principle of active substructure splicing, and the structures of these target compounds were confirmed by NMR and HRMS. In addition, the inhibitory activities of the resultant compounds against broadleaf and grass weeds were determined.

## 2. Results and Discussion

### 2.1. Chemistry

As can be seen from Figure 1 and Figure 2, all target compounds were obtained by multi-step reactions using substituted pyridines and ethyl 4,4,4-trifluoroacetoacetate as starting materials. Intermediates A (**4a**–**4e**) were prepared from substituted pyridine and p-hydroxyphenylboronic acid via multi-step reactions, such as the Suzuki cross-coupling reaction, nucleophilic substitution reaction, Newman–Kwart rearrangement reaction, and hydrolysis reaction [27,28]. Intermediate B was obtained via a simple three-step reaction as per the previously disclosed method, using ethyl 4,4,4-trifluoroacetoacetate as the starting material [29]. The target compounds **6a**–**6e** were prepared via a nucleophilic substitution reaction from intermediates A and B; compounds **6a**–**6e** were oxidized to yield compounds **7a**–**7e** using 3-chloroperbenzoic acid as the oxidant, according to previously disclosed methods [30]. After synthesis, all target compounds were characterized via HRMS and NMR. The NMR and HRMS spectra of all the target compounds are shown in the Appendix A.

### 2.2. Greenhouse Herbicidal Activity Assays

According to the herbicidal activity test results, none of the target compounds exhibited herbicidal activities for pre-emergence. As can be seen from Table 1, a few target compounds exhibited moderate herbicidal activities. Of these, at 150 g a.i./hm^2^, compounds **6a** and **6c** exhibited 50–60% inhibitory activities when used for the post-emergence treatment of the weeds *Digitaria sanguinalis* (*DS*), *Abutilon theophrasti* (*AT*), and *Eclipta prostrate* (*EP*). Furthermore, the inhibitory activities of compounds **6a** and **6c** against *EP* were superior to pyroxasulfone. 

From previous studies on the herbicidal activity of pyrazole derivatives, it can be seen that some reported pyrazole derivatives showed good herbicidal activity. According to the study of Zhou et al. [31], the herbicidal activity of some substituted phenylpyrazole derivatives against *Abutilon theophrasti*, at 150 g a.i./hm^2^, was above 90%. Although compounds **6a** and **6c** of this work exhibited moderate herbicidal activities, they could also be further optimized as lead compounds to obtain compounds with higher activity.

From Table 1, we can see that the herbicidal activities of compound **6** were obviously better than those of compound **7**, indicating that the structure containing 4-(pyridin-2-yl)phenylene sulfide was beneficial to the improvement of the activity. According to the SAR of compound **6** in the field of herbicidal activity, when the 3-position of pyridine was a chlorine atom and the 5-position was a fluorine atom or a trifluoromethyl group, compound **6** exhibited the best herbicidal activity for post-emergence.

## 3. Materials and Methods

### 3.1. Instrumentation

All reagents and other materials were purchased from commercial sources and used without additional purification unless otherwise noted. A B-545 melting point instrument (Buchi, Hangzhou, China) was used to determine the melting point without calibration. A Bruker AV-400 spectrometer (Billerica, MA, USA) was used to generate NMR spectra with DMSO-*d_6_* serving as the solvents. An Agilent 6545 Q-TOF LCMS spectrometer (Santa Clara, CA, USA) was used for mass spectrometry.

### 3.2. Synthesis 

The synthesis approach for pyrazole derivatives containing substituted 4-(pyridin-2-yl)benzene moieties in this work is outlined in Figure 1.

#### 3.2.1. Synthesis of Intermediates A (**4a**–**4e**)

4-(3-Chloro-5-trifluoromethylpyridin-2-yl) thiophenol intermediate (**4a**) is taken as an example. 

2,3-Dichloro-5-trifluoromethylpyridine (1.08 g, 5 mmol), potassium carbonate (1.38 g, 10 mmol), triphenylphosphorus (0.13 g, 10 mol%), p-hydroxybenzeneboronic acid (0.76 g, 5.5 mmol), palladium(II) acetate (5 mol%, 0.06 g), CH_3_OH (5 mL), and CH_3_CN (10 mL) were mixed and stirred at 50 °C for 6 h under N_2_. Thereafter, the mixture was extracted using ethyl acetate (30 mL×3), rinsed using brine, and concentrated. The remaining residue was then recrystallized using ethanol and water as solvents at 70 °C to obtain 1.22 g of compound **1a**. 

Compound **1a** (13.65 g, 50 mmol), 1,4-diazabicyclo[2.2.2]octane (11.22 g, 100 mmol), dimethylcarbamothioic chloride (11.22 g, 100 mmol), and *N*,*N*-dimethylformamide (250 mL) were mixed and stirred at 60 °C for 8 h. Thereafter, the mixture was extracted using ethyl acetate (100 mL × 3), rinsed using brine, and concentrated to give a yellow solid **2a**, which was used in the next reaction without further purification.

The yellow solid **2a** synthesized in the previous step and *N*-methyl pyrrolidone (100 mL) were stirred under reflux for 7 h under N_2_. Thereafter, the mixture was extracted using ethyl acetate (100 mL × 3), rinsed using brine, and concentrated to give a yellow solid **3a**, which was used in the next reaction without further purification. 

The yellow solid **3a** (3.61 g, 10 mmol), 85% potassium hydroxide (0.69 g, 10.5 mmol), tetrahydrofuran (20 mL), and methanol (10 mL) were mixed and stirred at 20 °C for 10 h. Thereafter, the mixture was made more acidic using hydrochloric acid, extracted thrice using ethyl acetate (30 mL×3), rinsed using brine, and concentrated. Residues were then purified via silica gel column chromatography using ethyl acetate (EA) and petroleum ether (PE) (V_EA_:V_PE_=1:10) to obtain 3.15 g of yellow solid of intermediate **4a**.

#### 3.2.2. Synthesis of Intermediate B

4-(Chloromethyl)-5-(difluoromethoxy)-1-methyl-3-(trifluoromethyl)-1H-pyrazole was prepared using the method disclosed previously, which is indicated in Figure 2.

#### 3.2.3. General Approach to the Synthesis of Compounds **6a**–**6e** and **7a**–**7e**

The target compounds **6a** and **7a** are taken as examples.

Compound **4a** (0.44 g, 1.5 mmol), 60% NaH (0.12 g, 3 mmol), and *N*,*N*-dimethylformamide (10 mL) were mixed and stirred at 20 °C for 30 min under N_2_. Next, 4-(chloromethyl)-5-(difluoromethoxy)-1-methyl-3-(trifluoromethyl)-1H-pyrazole **5** (0.48 g, 1.8 mmol) was added followed by stirring for 4 h at 60 °C. Thereafter, the mixture was extracted thrice using ethyl acetate (30 mL × 3), rinsed using brine, and concentrated. Residues were then purified via silica gel column chromatography using ethyl acetate (EA) and petroleum ether (PE) (V_EA_:V_PE_ = 1:10) to obtain 0.56 g of white solid of target compound **6a**.

Compound **6a** (0.20 g, 0.19 mmol) and 85% m-chloroperoxybenzoic acid (0.19 g, 0.93 mmol) in dichloromethane (5 mL) were mixed and stirred at 20 °C for 11 h. Thereafter, the mixture was evaporated to remove the solvent. Residues were then purified via silica gel column chromatography using ethyl acetate (EA) and petroleum ether (PE) (V_EA_:V_PE_=1:5) to obtain 0.07 g of white solid of target compound **7a**. 

3-chloro-2-(4-(((5-(difluoromethoxy)-1-methyl-3-(trifluoromethyl)-1H-pyrazol-4-yl)methyl)thio)phenyl)-5-(trifluoromethyl)pyridine (**6a**): White solid; Yield 72.1%; M.p. 68.3–70.4 °C. ^1^H NMR (400 MHz, DMSO-*d_6_*) *δ*: 9.04 (d, *J* = 0.9 Hz, 1H), 8.58 (d, *J* = 1.3 Hz, 1H), 7.73 (d, *J* = 8.4 Hz, 2H), 7.50 (d, *J* = 8.4 Hz, 2H), 7.31 (t, *J* = 71.5 Hz, 1H), 4.17 (s, 2H), 3.79 (s, 3H). ^13^C NMR (101 MHz, DMSO-*d_6_*) *δ*: 159.10, 144.95 (q, *J* = 3.5 Hz), 143.70 (t, *J* = 3.1 Hz), 138.24, 137.50 (q, *J* = 37.2 Hz), 136.34 (q, *J* = 3.5 Hz), 135.01, 130.46, 129.99, 128.33, 125.41 (q, *J* = 33.1 Hz), 123.32 (q, *J* = 273.9 Hz), 121.52 (q, *J* = 270.5 Hz), 116.78 (t, *J* = 268.4 Hz), 104.65, 36.24, 24.40. HRMS (ESI): calculated for C_19_H_13_ClF_8_N_3_OS [M+H]^+^ 518.0335 and found to be 518.0333.

3-bromo-5-chloro-2-(4-(((5-(difluoromethoxy)-1-methyl-3-(trifluoromethyl)-1H-pyrazol-4-yl)methyl)thio)phenyl)pyridine (**6b**): Yellow oil; Yield 75.7%; ^1^H NMR (400 MHz, DMSO-*d_6_*) *δ*: 8.73 (d, *J* = 2.1 Hz, 1H), 8.47 (d, *J* = 2.1 Hz, 1H), 7.63 − 7.60 (m, 2H), 7.48 − 7.44 (m, 2H), 7.30 (t, *J* = 71.6 Hz, 1H), 4.15 (s, 2H), 3.79 (s, 3H). ^13^C NMR (101 MHz, DMSO-*d_6_*) *δ*: 155.01, 146.95, 143.24 (t, *J* = 3.6 Hz), 140.58, 137.04 (q, *J* = 37.1 Hz), 136.75, 136.22, 129.94, 129.90, 128.03, 121.07 (q, *J* = 270.6 Hz), 119.05, 116.32 (t, *J* = 268.6 Hz), 104.31, 35.78, 24.13. HRMS (ESI): calculated for C_18_H_13_BrClF_5_N_3_OS [M+H]^+^ 527.9566 and found to be 527.9562.

3-chloro-2-(4-(((5-(difluoromethoxy)-1-methyl-3-(trifluoromethyl)-1H-pyrazol-4-yl)methyl)thio)phenyl)-5-fluoropyridine (**6c**): Yellow solid; Yield 67.1%; M.p. 74.3–76.6 °C. ^1^H NMR (400 MHz, DMSO-*d_6_*) *δ*: 8.70 (d, *J* = 2.6 Hz, 1H), 8.22 (dd, *J* = 8.5, 2.5 Hz, 1H), 7.65 − 7.61 (m, 2H), 7.48 − 7.43 (m, 2H), 7.30 (t, *J* = 71.5 Hz, 1H), 4.15 (s, 2H), 3.79 (s, 3H). ^13^C NMR (101 MHz, DMSO-*d_6_*) δ: 157.96 (d, *J* = 261.1 Hz), 151.96 (d, *J* = 4.0 Hz), 143.69 (t, *J* = 3.7 Hz), 137.48 (q, *J* = 37.1 Hz), 136.96 (d, *J* = 5.4 Hz), 136.71, 135.58, 130.35, 129.59 (d, *J* = 4.6 Hz), 128.67, 126.11 (d, *J* = 21.5 Hz), 121.53 (q, *J* = 270.5 Hz), 116.79 (t, *J* = 268.5 Hz), 104.80, 36.23, 24.63. HRMS (ESI): calculated for C_18_H_13_ClF_6_N_3_OS [M+H]^+^ 468.0367 and found to be 468.0368.

5-chloro-2-(4-(((5-(difluoromethoxy)-1-methyl-3-(trifluoromethyl)-1H-pyrazol-4-yl)methyl)thio)phenyl)-3-fluoropyridine (**6d**): Yellow solid; Yield 81.2%; M.p. 79.9–82.5 °C. ^1^H NMR (400 MHz, DMSO-*d_6_*) *δ*: 8.63 (s, 1H), 8.20 (d, *J* = 10.9 Hz, 1H), 7.88 (d, *J* = 7.7 Hz, 2H), 7.50 (d, *J* = 7.7 Hz, 2H), 7.31 (t, *J* = 71.4 Hz, 1H), 4.16 (s, 2H), 3.79 (s, 3H). ^13^C NMR (101 MHz, DMSO-*d_6_*) *δ*: 156.26 (d, *J* = 265.6 Hz), 144.42 (d, *J* = 4.6 Hz), 143.22, 142.85 (d, *J* = 10.3 Hz), 137.68, 137.03 (q, *J* = 37.3 Hz), 131.84 (d, *J* = 5.9 Hz), 129.96 (d, *J* = 3.8 Hz), 128.93 (d, *J* = 6.2 Hz), 128.42, 125.09 (d, *J* = 23.9 Hz), 121.07 (q, *J* = 270.3 Hz), 116.33 (t, *J* = 268.5 Hz), 104.28, 35.79, 23.98. HRMS (ESI): calculated for C_18_H_13_ClF_6_N_3_OS [M+H]^+^ 468.0367 and found to be 468.0366.

3-chloro-2-(4-(((5-(difluoromethoxy)-1-methyl-3-(trifluoromethyl)-1H-pyrazol-4-yl)methyl)thio)phenyl)-5-methylpyridine (**6e**): Yellow solid; Yield 79.0%; M.p. 106.4–108.7 °C. ^1^H NMR (400 MHz, DMSO-*d_6_*) *δ*: 8.47 (s, 1H), 7.89 (s, 1H), 7.64 (d, *J* = 8.3 Hz, 2H), 7.45 (d, *J* = 8.3 Hz, 2H), 7.30 (t, *J* = 71.5 Hz, 1H), 4.14 (s, 2H), 3.79 (s, 3H), 2.35 (s, 3H). ^13^C NMR (101 MHz, DMSO-*d_6_*) *δ*: 151.78, 148.43, 143.24 (t, *J* = 3.8 Hz), 138.44, 137.03 (q, *J* = 37.1 Hz), 136.08, 135.94, 134.06, 129.88, 128.48, 128.26, 121.08 (q, *J* = 270.6 Hz), 116.35 (t, *J* = 268.4 Hz), 104.41, 35.78, 24.27, 17.03. HRMS (ESI): calculated for C_19_H_16_ClF_5_N_3_OS [M+H]^+^ 464.0617 and found to be 464.0617.

3-chloro-2-(4-(((5-(difluoromethoxy)-1-methyl-3-(trifluoromethyl)-1H-pyrazol-4-yl)methyl)sulfonyl)phenyl)-5-(trifluoromethyl)pyridine (**7a**): White solid; Yield 34.4%; M.p. 95.5–98.0 °C. ^1^H NMR (400 MHz, DMSO-*d_6_*) *δ*: 9.11 (d, *J* = 1.1 Hz, 1H), 8.67 (d, *J* = 1.3 Hz, 1H), 8.01 − 7.93 (m, 4H), 7.26 (t, *J* = 71.8 Hz, 1H), 4.58 (s, 2H), 3.80 (s, 3H). ^13^C NMR (101 MHz, DMSO-*d_6_*) *δ*: 157.94, 144.85 (t, *J* = 4.1 Hz), 144.70 (q, *J* = 3.9 Hz), 142.25, 138.68, 138.14 (q, *J* = 37.2 Hz), 136.06 (q, *J* = 3.5 Hz), 130.43, 130.03, 128.16, 125.86 (q, *J* = 33.3 Hz), 122.76 (q, *J* = 274.2 Hz), 120.60 (q, *J* = 270.9 Hz), 116.63 (t, *J* = 267.7 Hz), 96.20, 49.50, 36.03. HRMS (ESI): calculated for C_19_H_13_ClF_8_N_3_O_3_S [M+H]^+^ 572.0052 and found to be 572.0050.

3-bromo-5-chloro-2-(4-(((5-(difluoromethoxy)-1-methyl-3-(trifluoromethyl)-1H-pyrazol-4-yl)methyl)sulfonyl)phenyl)pyridine (**7b**): White solid; Yield 49.0%; M.p. 136.9–139.4 °C. ^1^H NMR (400 MHz, DMSO-*d_6_*) *δ*: 8.80 (s, 1H), 8.56 (s, 1H), 7.90 (s, 4H), 7.25 (t, *J* = 71.8 Hz, 1H), 4.57 (s, 2H), 3.80 (s, 3H). ^13^C NMR (101 MHz, DMSO-*d_6_*) *δ*: 154.19, 147.21, 144.81 (t, *J* = 3.7 Hz), 143.78, 140.74, 138.18, 138.15 (q, *J* = 37.0 Hz), 130.92, 130.36, 128.02, 120.62 (q, *J* = 270.4 Hz), 119.28, 116.61 (t, *J* = 269.9 Hz), 96.22, 49.48, 36.05. HRMS (ESI): calculated for C_18_H_13_BrClF_5_N_3_O_3_S [M+H]^+^ 581.9284 and found to be 581.9286.

3-chloro-2-(4-(((5-(difluoromethoxy)-1-methyl-3-(trifluoromethyl)-1H-pyrazol-4-yl)methyl)sulfonyl)phenyl)-5-fluoropyridine (**7c**): White solid; Yield 95.0%; M.p. 106.8–109.2 °C. ^1^H NMR (400 MHz, DMSO-*d_6_*) *δ*: 8.76 (d, *J* = 2.5 Hz, 1H), 8.29 (dd, *J* = 8.5, 2.5 Hz, 1H), 7.91 (s, 4H), 7.24 (t, *J* = 71.8 Hz, 1H), 4.56 (s, 2H), 3.79 (s, 3H).^13^C NMR (101 MHz, DMSO-*d_6_*) *δ*: 158.46 (d, *J* = 262.0 Hz), 151.07 (d, *J* = 4.1 Hz), 145.29 (t, *J* = 3.6 Hz), 143.05, 138.60 (q, *J* = 37.0 Hz), 138.52, 137.18 (d, *J* = 23.1 Hz), 130.82, 130.05 (d, *J* = 5.0 Hz), 128.52, 126.28 (d, *J* = 21.6 Hz), 121.06 (q, *J* = 270.1 Hz), 117.08 (t, *J* = 267.6 Hz), 96.69, 49.97, 36.47.HRMS (ESI): calculated for C_18_H_12_ClF_6_N_3_O_3_SNa [M+Na]^+^ 522.0084 and found to be 522.0084.

5-chloro-2-(4-(((5-(difluoromethoxy)-1-methyl-3-(trifluoromethyl)-1H-pyrazol-4-yl)methyl)sulfonyl)phenyl)-3-fluoropyridine (**7d**): Yellow oil; Yield 85.0%; ^1^H NMR (400 MHz, DMSO-*d_6_*) *δ*: 8.72 (d, *J* = 1.1 Hz, 1H), 8.30 (dd, *J* = 11.0, 1.9 Hz, 1H), 8.16 (d, *J* = 7.4 Hz, 2H), 7.96 (d, *J* = 8.5 Hz, 2H), 7.26 (t, *J* = 71.8 Hz, 1H), 4.56 (s, 2H), 3.81 (s, 3H). ^13^C NMR (101 MHz, DMSO-*d_6_*) *δ*: 157.16 (d, *J* = 266.8 Hz), 145.30 (t, *J* = 4.6 Hz), 142.20 (d, *J* = 10.5 Hz), 139.78 (d, *J* = 5.8 Hz), 139.04, 138.60 (q, *J* = 37.2 Hz), 131.92 (d, *J* = 4.2 Hz), 129.86 (d, *J* = 6.2 Hz), 129.02, 125.96 (d, *J* = 23.9 Hz), 121.06 (q, *J* = 270.5 Hz), 117.07 (t, *J* = 267.3 Hz), 96.62, 49.97, 36.48. HRMS (ESI): calculated for C_18_H_12_ClF_6_N_3_O_3_SNa [M+Na]^+^ 522.0084 and found to be 522.0085.

3-chloro-2-(4-(((5-(difluoromethoxy)-1-methyl-3-(trifluoromethyl)-1H-pyrazol-4-yl)methyl)sulfonyl)phenyl)-5-methylpyridine (**7e**): White solid; Yield 80.7%; M.p. 116.9–119.8 °C.^1^H NMR (400 MHz, DMSO-*d_6_*) *δ*: 8.53 (s, 1H), 7.97 (s, 1H), 7.95 − 7.85 (m, 4H), 7.26 (t, *J* = 71.8 Hz, 1H), 4.56 (s, 2H), 3.80 (s, 3H), 2.39 (s, 3H). ^13^C NMR (101 MHz, DMSO-*d_6_*) *δ*: 151.36, 149.13, 145.29 (t, *J* = 3.2 Hz), 143.80, 139.03, 138.61 (q, *J* = 37.5 Hz), 138.24, 135.60, 130.75, 129.27, 128.42, 121.08 (q, *J* = 269.3 Hz), 117.10 (t, *J* = 267.5 Hz), 96.68, 49.97, 36.47, 17.57. HRMS (ESI): calculated for C_19_H_15_ClF_5_N_3_O_3_SNa [M+Na]^+^ 518.0335 and found to be 518.0337.

### 3.3. Herbicidal Activity Test

Levels of herbicidal activity for compounds **6a**–**6e** and **7a**–**7e** against the monocotyledonous weeds *Digitaria sanguinalis (DS)*, *Echinochloa crusgalli (EC)*, and *Setaria viridis (SV)*, and the dicotyledonous weeds *Abutilon theophrasti (AT)*, *Amaranthus retroflexus (AR)*, and *Eclipta prostrate (EP)* were determined using previously disclosed methods [32], with the results being listed in Table 1.

## 4. Conclusions

In conclusion, 10 novel pyrazole derivatives containing phenylpyridine moieties were prepared using pyroxasulfone as the lead compound. Among these, compounds **6a** and **6c** possessed moderate activity (50%) against *EP* for post-emergence at 150 g a.i./hm^2^, which was slightly superior to pyroxasulfone. This study suggested that it may be the introduction of the phenylpyridine structure that allowed the target compounds to exhibit herbicidal activity at post-emergence only. Thus, compounds **6a** and **6c** may be lead compounds for further structural optimization.

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
