# Peer review of "Synthesis of Novel Pyrazole Derivatives Containing Phenylpyridine Moieties with Herbicidal Activity"

_molecules, 2022, doi:10.3390/molecules27196274_

Round 1

Reviewer 1 Report

The present manuscript describes synthesis of novel heterocyclic compounds and their herbicidal properties. It is definitely of high practical importance. In my opinion, the manuscript can be accepted by Molecules after minor revision:

1. 1H NMR of 6b: unlabeled signal at 5.75 – what is this? 13C NMR of 6b: unlabeled signal at 55: what is this? Where did it come from?

2. 1H NMR of 7a: unlabeled signals at 7.50-7.75 – what is this? 13C NMR of 7a: unlabeled signals at 133, 129, 128: what is this? Where did it come from? In compounds 7c, 7d, a similar impurity is observed.

3. Calculated HRMS values should be thoroughly checked.

Reviewer 2 Report

The manuscript  by Xiaohua Du entitled Synthesis of Novel Pyrazole Derivatives Containing 2 Phenylpyridine Moieties with Herbicidal Activity. The bioassay revealed that a few com- 13 pounds exhibited moderate herbicidal activities against Digitaria sanguinalis, Abutilon theophrasti, 14 and Setaria viridis for post-emergence treatment. For instance, compounds 6a and 6c demonstrated 50% inhibition activity against Setaria viridis, which was slightly superior to pyroxasulfone.This study demonstrated that the target 221 compounds only exhibited herbicidal activity for post-emergence due to the introduction 222 of the phenylpyridine structure. Thus, compounds 6a and 6c may be lead compounds for 223 the further structural optimization.

The work done carefully, well written and the NMR of the compounds  reported in this manuscript are high quality with clear base lines and it deserves the publication in the Molecules.

Author Response

Thank you for acknowledging the article.

Reviewer 3 Report

Authors described the synthesis of novel pyrazole derivatives containing phenylpyridine moieties with herbicidal activity. I consider that the manuscript does not meet all requirements to be published in “Molecules” (Q-1 in SCImago) due to the little information. The explanation of all sections are very superficial. Additional suggestions and comments are included:   (1) See introduction. It does have enough scientific rigor. It is very important to include previous studies on pyrazole derivatives with herbicidal activity. (2) See 2.1. Chemistry. It is very important to include yield of all products in schemes 1 and 2.  The compounds 6a-e have the better herbicidal activity. I consider that the scope should be extended and complemented with docking molecular studies. (3) See 2.2. Greenhouse herbicidal activity assays. The synthesized compounds exhibited low herbicidal activities. These results should be complemented by docking molecular studies. (4) See 3.2. Synthesis. NMR and HRMS reporting data of all synthesized compounds should be included. (5) See conclusion. It should be re-structured. (6) See references. The DOI of all articles should be added. (7) NMR and HRMS spectra of “all” synthesized compounds should be included in the Supplementary Material.  

Round 2

Reviewer 3 Report

The authors described the synthesis of novel pyrazole derivatives containing phenyl-pyridine moieties with herbicidal activity. The authors included NMR and HRMS spectra and reported data of all synthesized compounds. However, I consider that introduction and explanation of herbicidal activities might be improved. Additional suggestions and comments are included:   (1) See introduction. It does have enough scientific rigor. It is very important to include previous studies on pyrazole derivatives with herbicidal activity including the most relevant herbicidal activities and previous synthetic methods.   (2) See 2.2. Greenhouse herbicidal activity assays. The synthesized compounds exhibited low herbicidal activities. These results should be complemented by docking molecular studies. If not, the discussion should be focused on previous reporting data of pyrazole analogs.   (3) See references. The DOI of all articles should be added.

Author Response

Point 1: See introduction. It does have enough scientific rigor. It is very important to include previous studies on pyrazole derivatives with herbicidal activity including the most relevant herbicidal activities and previous synthetic methods.

Response 1: We have added the previous studies on pyrazole derivatives with herbicidal activity in the introduction section.

Point 2: See 2.2. Greenhouse herbicidal activity assays. The synthesized compounds exhibited low herbicidal activities. These results should be complemented by docking molecular studies. If not, the discussion should be focused on previous reporting data of pyrazole analogs.

Response 2: We have tried molecular docking, but no suitable enzyme has been screened, so we discussed previous reporting data of pyrazole analogs.

Point 3: See references. The DOI of all articles should be added.

Response 3: We have added the DOI of all articles in the references section.